# Absence of Seed-Mediated Transmission of Cucumber Mosaic Virus in Espelette Pepper Crops despite Widespread and Recurrent Epidemics

**DOI:** 10.3390/v15112159

**Published:** 2023-10-27

**Authors:** Judith Hirsch, Marion Szadkowski, Catherine Wipf-Scheibel, Elise Lepage, Paul Dumeaux, Elodie Nguyen, Eric Verdin, Benoît Moury, Loup Rimbaud

**Affiliations:** 1INRAE, Pathologie Végétale, 84140 Avignon, France; marion.szadkowski@inrae.fr (M.S.); catherine.rys@inrae.fr (C.W.-S.); elise.lepage@inrae.fr (E.L.); eric.verdin@inrae.fr (E.V.); benoit.moury@inrae.fr (B.M.); 2AgroParisTech, 91123 Palaiseau, France; 3INRAE, BioSP, 84140 Avignon, France; 4Syndicat du Piment d’Espelette AOP, 64250 Espelette, France; paul.dumeaux@pimentdespelette.com (P.D.); elodie.nguyen87@gmail.com (E.N.)

**Keywords:** cucumber mosaic virus, pepper, *Capsicum annuum*, vertical transmission, seed transmission

## Abstract

In the past decade, severe epidemics of cucumber mosaic virus (CMV) have caused significant damage to Espelette pepper crops. This virus threatens the production of Espelette pepper, which plays a significant role in the local economy and touristic attractiveness of the French Basque Country, located in southwestern France. In 2021 and 2022, CMV was detected via double-antibody sandwich enzyme-linked immunosorbent assays (DAS-ELISAs) in Gorria pepper seed lots harvested from naturally infected fields scattered throughout the entire Espelette pepper production area. These seed lots were used in greenhouse grow-out tests to determine whether CMV could be transmitted to seedlings from contaminated seeds, using visual symptom assessment, DAS-ELISAs, and reverse transcription-polymerase chain reaction (RT-PCR). Despite the widespread occurrence of CMV in seeds of field samples, the grow-out experiments on a total of over 5000 seedlings yielded no evidence of seed transmission of local CMV isolates in Gorria pepper. Therefore, rather than seeds from infected pepper plants, sources of CMV inoculum in Espelette are more likely to be alternative hosts present in and around pepper fields that can allow for the survival of CMV during the off-season. These results have important epidemiological implications and will guide the choice of effective measures to control current epidemics.

## 1. Introduction

With more than 5 million plants every year in a 300 ha zone, Espelette pepper is a product emblematic of the French Basque Country, located in the southwest of France. Espelette pepper products, mainly ground chili powder and pepper strings that were originally dried on the façades of Basque houses, are world-renowned and foster a flourishing tourist industry, with over 300,000 visitors per year to the village of Espelette.

Espelette pepper has been granted the French label “Appellation d’origine contrôlée” (AOC) since 2000, and the European Union Protected Designation of Origin (PDO) since 2002. This designation, which is a type of geographical indication, means that growers, processors, and producers must comply with a number of specifications listed in the Book of Specifications. In particular, seeds of the local *Capsicum annuum* L. cultivar “Gorria” (meaning “red” in Basque) are produced exclusively by farmers (about 200 in total) through mass selection, i.e., seeds from individual plants selected on the basis of phenotype (mainly precocity, fruit morphology, and quality) are bulked and used to grow the next-generation plants. Seeds are sown by each farmer under the cover of greenhouses (mainly in February and March, and occasionally in April), followed by transplanting in open fields (May to 15 July).

Despite a highly fragmented landscape (small pepper fields, surrounded by fields of other crops such as corn, and hilly terrains) and the use by farmers of their own locally selected seeds, in the past decade cucumber mosaic virus (CMV, genus *Cucumovirus*, family *Bromoviridae*) epidemics have become widespread in the PDO area, causing recurrent damage to Espelette pepper crops and becoming a major threat to Espelette pepper production. CMV, a single-stranded positive-sense RNA virus, has a host range of over 1000 plant species, including many crops and wild species [1]. In pepper crops, CMV infection can lead to more or less severe stunting of infected plants and reduced fruit yield and quality. Foliar symptoms of CMV infection include leaf mottling, mosaics, vein-clearing, filiformism (the “shoestring” appearance of leaf blades), and typical “oak-leaf” necrotic lesions on older leaves. Infected fruits can present typical concentric rings, necrotic spots, roughness, and deformations. CMV is transmitted horizontally by over 80 species of aphids in a non-persistent non-circulative manner, a mode of virus transmission characterized by short acquisition and inoculation times (from seconds to minutes) via aphid vectors whose stylets become contaminated during the probing of infected plants. The vertical (seed) transmission of CMV has also been reported in over 40 host species belonging to a wide range of botanical families, with efficiencies that vary from less than 0.1% in certain host species like zucchini, to 100% in certain soybean cultivars [2,3].

In order to design appropriate disease management strategies with which to control these epidemics, it is essential to acquire data on the epidemiology of this viral disease in the local context and understand how CMV is transmitted. In particular, it is currently unclear how CMV re-emerges every year on pepper crops despite the absence of peppers during the winter months. Is the virus transmitted horizontally via aphid vectors from infected neighboring crop plants or weeds, and/or vertically through infected pepper seeds? Are both routes important for dissemination of the virus in Espelette?

Based on the results of industry grow-out tests performed on commercial pepper seed lots, the seed transmission of CMV in pepper appears to be an unlikely route for the propagation of the virus (International Seed Federation (ISF)-regulated pest list database [4]); however, there have been two recent reports of vertical transmission of CMV in pepper, both in experimental conditions and from naturally infected seeds [5,6]. In both cases, high transmission rates were observed: 10–14% for the Fny isolate in mechanically inoculated plants of the *C. annuum* cultivar Marengo, and as high as 57–86% in three naturally infected *Capsicum* spp. accessions.

This study addresses the question of the seed transmission of CMV in the local “Gorria” pepper cultivar through a series of greenhouse grow-out tests. Seedlings from naturally infected seed lots were scored for symptoms and screened for the presence of CMV in leaves via the use of serological and molecular assays.

## 2. Materials and Methods

Two distinct experiments were carried out to evaluate the occurrence and efficiency of the vertical transmission of local CMV isolates in the Espelette pepper cultivar, called “Gorria”. These experiments differed by how the pepper seeds were collected.

### 2.1. Pepper Seeds in Grow-Out Experiment One

Seeds were sampled among bulks of seeds collected by pepper growers in 2020 and stored for sowing in the spring of 2021, except for one seed lot harvested in 2019. Every year during the harvest period farmers select seeds in their respective fields from plants bearing apparently healthy fruits, chosen for their phenotypic performance (precocity, fruit quantity, and quality). This yielded 22 seed lots from 21 producers throughout the Espelette production basin, each using their own locally selected seeds. No disinfection treatments were applied to the seeds (except for one seed lot which was treated with bleach by the farmer).

### 2.2. Pepper Seeds in Grow-Out Experiment Two

Seeds were extracted from pepper fruits collected from symptomatic Gorria pepper plants in Espelette pepper fields in September 2021. The symptomatic parent plants were infected with CMV, as ascertained by double-antibody sandwich enzyme-linked immunosorbent assays (DAS-ELISAs) performed on symptomatic leaves. As for grow-out experiment one, no disinfection treatments were applied to the seeds.

### 2.3. Negative and Positive Controls

Leaves or seeds from uninfected plants of the Yolo Wonder (YW) cultivar were used as negative controls for DAS-ELISAs and reverse transcription-polymerase chain reactions (RT-PCRs). YW, *Nicotiana tabacum* cultivar Xanthi, or *Nicotiana benthamiana* plants infected with a CMV isolate belonging to the most frequent CMV haplotype detected in the Espelette area on pepper crops (GenBank accession numbers OR355463, OR355464, and OR355465 for RNAs 1 to 3, respectively) and/or the CMV I17F strain [7] were used as CMV-positive controls for DAS-ELISAs and RT-PCR assays. These negative and positive control plants were placed in separate greenhouse compartments to avoid cross-contamination. Additional healthy YW plants (24 in grow-out test one and 67 in grow-out test two) were placed at regular intervals in the greenhouse where the grow-out trials were performed as a control for the cross-contamination of samples during plant growth, harvesting, the grinding of samples, or RNA extraction.

### 2.4. Virus Detection in Seeds

Seed lots in both of the experiments were tested via DAS-ELISAs for the presence of CMV prior to sowing, and only infected seeds from lots containing more than 20 seeds were sown. To carry out these tests, seed samples were ground in Universal U-form extraction bags (BIOREBA AG, Reinach, Switzerland) as a precaution to avoid cross-contaminations between samples. After a pre-soaking step of 4 h on ice in 2 mL of a phosphate extraction buffer (0.03 M sodium phosphate (Na_2_HPO_4_) solution containing 0.2% diethyldithiocarbamate (DIECA) *w*/*v*), seed samples were first crushed coarsely by using a pressure press to avoid piercing the extraction bags, after which they were finely ground by using a home-developed homogenizer with a ball-bearing head.

For the seed lots used in grow-out experiment one, three batches of 30 seeds per seed lot were tested. For the seed lots used in grow-out experiment 2, where each seed lot was the progeny of a single pepper fruit, a batch of 3 to 10 seeds per seed lot was tested. The ground samples were analyzed via DAS-ELISAs (see below for details).

### 2.5. Grow-Out Experiment Conditions

The grow-out experiments were performed in a naturally illuminated greenhouse at INRAE Avignon, France, in the springs of 2021 and 2022. The greenhouse was heated and equipped with an adiabatic cooling system with a temperature set point of 25 °C. The average day/night temperatures were 25/20 °C for both of the grow-out tests. Seeds were directly sown in trays containing a peat and clay substrate (TS 3-404; Klasmann-Deilmann France SARL, Bourgoin Jallieu, France). For fertilization, similar amounts of N/P/K were applied via daily watering with a nutrient solution for both experiments. The N-P-K and MgO percentages were 16-5-25 and 2 in 2021 (SOLUVEG ESSENTIEL, Angibaud, La Rochelle, France), and 13-5-27 and 4 in 2022 (SOLUVEG ALC 57).

The overall scheme of the two seed transmission experiments is presented in Figure 1.

#### 2.5.1. Grow-Out Test One

Grow-out test one was conducted in April–May 2021. One hundred and twenty seeds per seed lot were sown in five trays containing 24 seeds each. The five trays per seed lot were placed in different locations in the greenhouse on five different benches (one tray per bench, with the twenty-two trays corresponding to the twenty-two seed lots distributed randomly along each of the five benches). Symptom scoring was performed weekly for six weeks (last symptom observation six weeks after sowing, shortly before plants were disposed of). Sampling for DAS-ELISA and RT-PCR tests was performed five weeks after sowing, when plants had reached the three- to four-leaf stage. The uppermost fully expanded leaf (3rd or 4th leaf) from each plant was sampled. DAS-ELISA tests were performed on batches of one to four seedlings (depending on the number of plants that germinated in each row of each tray). Equal volumes of homogenate from the six batches corresponding to one tray (i.e., 24 seedlings or less, depending on the germination rate of the seed lots) were pooled for RNA extraction and RT-PCR tests.

#### 2.5.2. Grow-Out Test Two

Grow-out test two was performed in April–June 2022. Forty seeds per CMV-positive fruit were sown in two trays containing twenty seeds each. For each of the 106 parental fruits, the two trays were placed randomly along two benches of the greenhouse. Plants were examined visually every week for the appearance of disease symptoms (last symptom observation seven weeks after sowing). Sampling and DAS-ELISA tests were performed four weeks after sowing, at approximately the same 3- to 4-leaf stage as for grow-out test one. As for grow-out test one, DAS-ELISA tests were performed on batches of one to four seedlings. RNA extractions and RT-PCR tests were performed on pools of 10 homogenates (corresponding to the 40 or fewer seedlings from a single CMV-positive fruit). To further explore doubtful RT-PCR results obtained from pooled samples, a subset of samples was individually tested using ELISAs and RT-PCR assays at 9 to 11 weeks after sowing.

### 2.6. Serological and Molecular CMV Detection Assays

#### 2.6.1. DAS-ELISAs

DAS-ELISAs [8] were performed with antisera provided by J. C. Devergne (INRA, Antibes) and directed against the D strain of CMV [9].

Seeds (batches of 10 or 30) were ground as described above. Leaf samples (1 g of 1 to 4 leaves) were ground in 4 mL of cold phosphate extraction buffer. The leaf samples were ground in a roller mill with a decontamination step using a 20% alkaline detergent (RBS T105; Traitements chimiques de surfaces S.A.R.L., Frelinghien, France) followed by a thorough rinsing with 100 mL of purified water (obtained via reverse osmosis) between samples. For seed samples, non-diluted homogenate was used directly for DAS-ELISA tests. For leaf samples, 30 µL of homogenate were diluted in 120 µL of phosphate extraction buffer and used for the ELISA tests.

As a precaution to avoid cross-contaminations between samples, after the sample loading step the ELISA plate wells were emptied with a vacuum pump, rather than by tapping the ELISA plates over the sink, as is sometimes done with other less concentrated viruses or highly diluted samples.

Positive, negative, and blank (buffer) controls were included in duplicate (or more) on all ELISA plates. For each ELISA plate, samples with absorbances measured at 405 nm (A_405_) above a threshold of 3 times the mean A_405_ of negative controls (seeds or leaves collected from non-infected plants of the cultivar Yolo Wonder) were considered positive. Samples with A_405_ below a threshold of twice the mean A_405_ of negative controls were considered negative. Samples with A_405_ between these two thresholds were considered “doubtful”.

#### 2.6.2. RT-PCR Analyses

Total RNAs were extracted from 200 µL of leaf homogenate using TRI Reagent (Molecular Research Center, Inc., Cincinnati, OH, USA) according to the manufacturer’s instructions. RNAs were resuspended in 20 µL of RNAse-free sterile distilled water. A two- step RT-PCR was performed on the RNA samples. Primer pairs used to amplify CMV sequences from genomic RNAs 1 and 3 are described in [10]. These primer pairs amplify PCR products of 376 and 436 base pairs (bps) in the helicase (1a) and coat protein (CP) genes, respectively.

Prior to the RT step, the RNA template and RT primer (PCR reverse primer) were mixed in a tube, denatured for 3 min at 80 °C, and cooled on ice for 5 min. The RT step was performed for 1 h at 42 °C on 2 µL of denatured total RNA in a total reaction volume of 20 µL using 4 units of AMV reverse transcriptase (Promega, Madison, WI, USA).

The PCR amplifications were carried out on 2 µL of viral cDNA in a total reaction volume of 20 µL using 1.5 units of Go*Taq* DNA polymerase (Promega) in an Agilent SureCycler 8800 thermocycler. The PCR conditions were as follows: 94 °C for 3 min for initial denaturation, followed by 40 cycles of denaturation at 94 °C for 30 s, annealing at 54 °C for 30 s, extension at 72 °C for 30 s, and a final extension at 72 °C for 10 min.

PCR products (4 or 5 µL) were analyzed via electrophoresis on a 1.5% agarose gel in a Tris-borate-EDTA (TBE) buffer stained with ethidium bromide, and by using a 1 kb DNA ladder (Promega). Positive controls (cDNA from leaf tissues infected with CMV) and negative controls (water) were included in all PCR runs.

#### 2.6.3. DAS-ELISA and RT-PCR Sensitivity Tests

Sensitivity tests were performed on our DAS-ELISA and RT-PCR procedures to ensure that a positive sample mixed in a batch of 1 to 4 and 1 to 40 healthy plants would invariably yield a positive result in DAS-ELISAs and RT-PCR, respectively.

Ten Gorria pepper plants were infected mechanically with a field-isolated CMV originating from Espelette. A crude extract from the CMV-infected Gorria pepper leaves was diluted *x*-fold in a crude extract from healthy Gorria leaves to reflect pools used for ELISA tests (*x* = 4) and RT-PCR tests (*x* = 24 for grow-out test one and *x* = 40 for grow-out test two). Higher dilutions were also used to reflect possibly very low virus titers in leaf tissues infected through vertical transmission (*x* = 100, *x* = 400, and *x* = 4000).

These control samples were analyzed using DAS-ELISAs and RT-PCR to assess the dilution factor up to which CMV can still be detected.

#### 2.6.4. Sanger Sequencing of PCR Products

For some doubtful results in grow-out test two, where PCR amplicons close to the expected size were obtained, the PCR products were sequenced using RT-PCR primers corresponding to CMV RNA 1. Sequencing was performed by GenoScreen (Lille, France).

## 3. Results

### 3.1. Detection of CMV in Gorria Pepper Seed Lots

In order to identify Gorria pepper seeds contaminated with the local CMV isolates to be tested in grow-out trials, DAS-ELISA tests were performed on two types of seed samples. The first samples consisted of seeds collected in 2019–2020, destined to be sown in 2021 by Espelette pepper producers cultivating pepper fields distributed across the PDO area (designated hereafter as the “farmer seed lots”). The ELISA tests revealed that all 22 seed lots provided by 21 pepper growers were positive for CMV (Figure 1 and Figure 2). These 22 CMV-positive seed lots were used for grow-out test one.

The second type of seeds (the “individual fruit seed lots”) were extracted from pepper fruits collected on CMV-infected plants during a field survey in the fall of 2021. Two hundred and thirty pepper fruits were collected from symptomatic plants in 23 fields cultivated by 23 different Espelette pepper growers (10 pepper fruits per field). These 23 pepper growers included the 21 growers who had provided the 2020 seed lots.

Among the 230 collected fruits, 217 yielded enough seeds to be tested with DAS-ELISAs. Seed lots from 106 fruits (49% of tested fruits) were CMV-positive and were used in grow-out test two (Figure 1 and Figure 2). Apart from one field for which there were no seed lots that were clearly positive for CMV, between 1 and 10 CMV-positive fruits (seed lots) were tested per field.

### 3.2. DAS-ELISA and RT-PCR Sensitivity Assays

Initial DAS-ELISA and RT-PCR sensitivity tests using control bulked samples were carried out on ten different infected Gorria pepper plants for DAS-ELISAs and five for RT-PCR, using different dilution factors. This experiment showed that leaf extracts from infected Gorria seedlings diluted four-fold in crude healthy leaf extracts systematically led to positive results in a DAS-ELISA test. In the RT-PCR test, our samples were consistently diagnosed as positive even when diluted as high as 400-fold in crude healthy leaf extracts (Figure 3 and Appendix A). This ensured that working with bulked samples did not preclude the detection of a single CMV-infected leaf mixed with up to three healthy leaves, which is the pooling strategy chosen for DAS-ELISA tests, or 23 or 39 healthy leaves for RT-PCR tests for grow-out experiments one and two, respectively.

### 3.3. Grow-Out Test One

In order to determine whether local CMV isolates could be vertically transmitted in Gorria pepper, we first screened the progeny of the “farmer” seed lots. A total of 2640 seeds were sown (Figure 1). The mean germination rate of the 22 seed lots was approximately 85% (Appendix A). This is in accordance with the average germination rate observed in the field for seeds of the Gorria cultivar. None of the 2247 seedlings that germinated displayed any typical CMV disease symptoms (Figure 1).

The leaves of the seedlings were screened for the presence of CMV. ELISA tests were performed on the uppermost, fully expanded leaves of the seedlings at 5 weeks after sowing using bulked samples of four leaves or less. None of the ELISA tests were positive (Figure 1 and Figure 2). In addition, a more sensitive RT-PCR assay was used to check for the eventual presence of CMV in the leaves of the seedlings at very low titers that would have been undetectable with ELISAs. The RT-PCR tests using a primer pair amplifying a 436 nt sequence from CMV genomic RNA 3 confirmed the ELISA tests: no PCR products were amplified from any of the bulked leaf samples tested (Figure 1 and Appendix A). Therefore, none of the leaf samples contained CMV, even at very low levels.

The absence of CMV-infected seedlings in the progeny of the 22 CMV-infected seed lots was thus confirmed through both serological and molecular methods.

### 3.4. Grow-Out Test Two

For the second grow-out test, the CMV-positive “individual fruit” seed lots extracted from fruits collected in the fall of 2021 from symptomatic plants were used. Of the 4167 seeds sown, 3069 germinated (73.7%). The germination rate varied between 0% and 100% between seed lots (Appendix A). The progeny seedlings were inspected visually every week for virus symptoms. None of the seedlings displayed any typical CMV symptoms during the course of the grow-out trial (Figure 1).

The progeny of each seed lot was assessed for the presence of CMV in bulked leaf samples (four leaves from four different seedlings per bulk) four weeks after sowing. The DAS-ELISA tests revealed no positive samples among the 961 bulks (Figure 1 and Figure 2).

RT-PCR experiments using a primer pair amplifying a 376 nt sequence from CMV genomic RNA 1 did not yield PCR products of the expected size for 99 out of the 103 bulked leaf samples; however, one sample displayed a strong PCR band of approximately the expected size, and three other samples displayed a faint band of the same size. These four samples had been handled side-by-side and processed simultaneously (Appendix A).

Using the same RNAs extracted from the four bulks, as well as negative and positive controls, new cDNA synthesis and PCR steps were performed and yielded the same results. Sequencing of the PCR products showed that the amplicon corresponded to the expected region on CMV genomic RNA 1 for three of the samples and did not yield an exploitable sequence for the fourth sample.

Were these four bulked leaf samples bona fide CMV-positive samples? Although the RT-PCR tests produced PCR amplicons, the DAS-ELISA tests were negative for these samples (see above and Figure 2). To rule out a possible cross-contamination of the four bulks during the sampling or grinding of leaves, the pooling of homogenates, or at the RNA extraction step, new leaf samples were collected on all individual seedlings of the four corresponding progenies, nine weeks after sowing. The DAS-ELISA tests performed on 113 individual leaves did not reveal the presence of CMV in any of the samples. In addition, a third series of leaf samples was collected from the same seedlings eleven weeks after sowing in order to perform RT-PCR tests. RT-PCRs using primers for CMV genomic RNAs 1 and 3 performed on 89 individual leaf samples from three of the seed lots and two freshly prepared bulked leaf samples from the fourth seed lot produced no PCR products, whereas positive controls yielded a PCR product of the expected size (Appendix A). Therefore, the testing of individual seedlings with ELISAs and RT-PCR revealed that the PCR products detected initially most likely resulted from the cross-contamination of the initial bulked samples (with positive controls) at the leaf sampling, grinding, pooling, or RNA extraction step.

Thus, the results from grow-out test two do not support the transmission of CMV from naturally contaminated seeds to seedlings in Gorria pepper.

## 4. Discussion

Our study, using large-scale grow-out tests coupled with serological and molecular CMV detection methods, yielded no evidence of the vertical transmission of CMV in Espelette Gorria pepper. No typical CMV symptoms were observed during the course of both seed transmission tests, nor was CMV detected in leaves of grown-out seedlings using either DAS-ELISA or RT-PCR detection methods. Overall, the two independent seedling grow-out experiments do not support the seed transmission of cucumber mosaic virus in the Gorria pepper cultivar from seeds naturally contaminated with CMV.

### 4.1. Could We Have Missed Very Low Virus Titers or Transmission Rates?

In our laboratory tests, the mechanical inoculation of local CMV isolates on Gorria seedlings produces clear, typical CMV symptoms that appear in systemic leaves within 6 to 8 days after inoculation. Consequently, seedlings which would be vertically infected by Espelette CMV isolates could also be expected to display disease symptoms. This was not the case, as none of the seedlings in the two transmission tests showed any symptoms, supporting the idea that there was no seed transmission of the virus in these experiments.

However, as CMV-infected plants can be asymptomatic, particularly when the virus is seed-transmitted [11,12], and vertically transmitted viruses have been shown to sometimes display very low titers in plant tissues [13,14,15], the detection of CMV-infected seedlings was not limited to screening for viral symptoms in our grow-out tests. Both ELISAs and RT-PCR assays were used to confirm the absence of the virus in the progeny of CMV-contaminated seeds. By using the highly sensitive RT-PCR assay, we have maximized our chances of detecting the virus, even if it was present at very low levels in the leaves of the seedlings. Indeed, in our laboratory experiments, the ELISA and RT-PCR sensitivity assays showed that CMV can be reliably detected in infected pepper leaves, even when samples were diluted 4- or 400-fold, respectively, in homogenate from healthy leaves (Figure 3).

In addition to the use of a highly sensitive CMV detection assay, our experimental setup was designed to enable the detection of very low vertical transmission rates. Over 5000 seedlings (2247 + 3069 = 5316 in the two vertical transmission tests; see Figure 1) were tested in the two grow-out tests, and no CMV-infected seedling was detected. Testing over 5000 seedlings can ensure > 95% probability that transmission rates as low as 0.06% are detected (based on the binomial distribution: 1 − (1 − 0.0006)^5000^ = 0.95).

### 4.2. Why Was CMV Detected in Gorria Pepper Seeds but Not in the Progeny of These Virus-Infected Seeds?

CMV was present in Espelette pepper seeds, most probably on the seed coat or other tissues of maternal origin, or possibly in the embryo, but this is less probable. The seed transmission of plant viruses can occur via different mechanisms. In the case of highly stable viruses like tobamoviruses, sobemoviruses, tritimoviruses, and poaceviruses, viruses present in or on the seed coat or other tissues of maternal origin can be transmitted to the seedling through wounds that occur during the germination, transplanting, or handling of seedlings [16,17,18]. For other less stable viruses, such as CMV, the most frequent mechanism of seed transmission relies on infection of the embryo. Infection of the embryo can occur through direct invasion via the suspensor that connects the parent plant to the developing seed and/or indirect invasion via infection of the male and/or female gametophyte(s), i.e., pollen and/or ovules [3,19,20]. The aim of this study was to determine whether CMV can be seed-transmitted and to deduce practical control measures, and not to elucidate the mechanisms of seed transmission. Therefore, although very interesting from a fundamental point of view, the question of the localization of the virus on or in pepper seeds was not further investigated.

Despite being detectable using DAS-ELISAs in batches of Espelette pepper seeds tested within the 3 to 10 months following their harvest (and even over 18 months after harvest in the case of the single 2019 “farmer” seed lot tested in 2021), CMV was not transmitted vertically under our experimental conditions, as we did not find any CMV-infected seedlings in either of the two grow-out tests. Since ELISA tests detect both infectious and non-infectious virus particles, detection of the virus using this serological method in the seeds does not mean that the virus is still infectious. CMV is a highly unstable virus, and although CMV virions can survive for long periods within seeds of certain host species (at least one year in *Cucurbita pepo* seeds and 27 months in *Vigna unguiculata* [3]), the infectivity of the virus may not be preserved as long in Espelette pepper seeds. Moreover, the virus may be degraded during germination [5].

### 4.3. Why Do Our Results Differ from the Two Published Reports of CMV Vertical Transmission in Pepper?

Data from experiments with mechanically infected parent plants suggest that seed can be a pathway for CMV transmission for certain CMV isolates, at least in certain pepper accessions [5,6]. Moreover, there is additional evidence for CMV seed transmission from naturally infected seeds in *Capsicum* spp. [6].

The discrepancy between our results and these two reports can be easily explained, since transmission efficiency can vary greatly depending on many factors, including the particular plant cultivar and virus isolate combination, the phenological stage at the time of infection, and the environmental conditions [3,14,19]. In the first study by Ali and Kobayashi [5], high levels of seed transmission (10–14%) were reported in experimental conditions using one pepper cultivar, Marengo, and one CMV strain, the Fny strain. In the second study, Arogundade et al. [6] reported very high transmission rates (57–86%), which varied depending on the *Capsicum* spp. cultivar.

CMV strains can be classified into two major genetic groups, I and II, with nucleotide identity between these two groups varying between 69 and 77%. Group I strains can be further subdivided into subgroups IA and IB [21]. Subgroup IA strains have a worldwide distribution, while most subgroup IB strains are from East Asia, the presumed geographic origin of this subgroup [2]. Unlike the Fny strain, which belongs to subgroup IA, “Espelette” pepper CMV isolates belong to subgroup IB [22]. No information is available concerning the classification of the CMV isolates from Nigeria studied by Arogundade et al. [6]. The genetic distance between these different CMV strains may account in part for the differences observed in seed transmission efficiency in pepper.

Altogether, our study and these two studies show that CMV seed transmission in pepper can occur only for certain CMV strains or pepper cultivars or only under certain environmental conditions, and that, when seed transmission does occur, the transmission rates can vary considerably.

### 4.4. Why Was CMV Present in All Seed Lots Derived from Farmer Mass Selection?

All of the “farmer” seed lots collected in 2020 by Espelette pepper producers were positive for CMV, as assessed with DAS-ELISA tests. Although this result was consistent with the observation that CMV epidemics were widespread in the Espelette area in 2020, it was somewhat surprising. Indeed, since the seeds were derived from farmer mass selection, we had not expected them to be massively infected with CMV. This result raises several questions worth exploring to better understand the expression of symptoms in CMV-infected pepper fruits versus vegetative organs. The seeds were extracted from the “best” Gorria pepper fruits selected by farmers to be sown the following season. If CMV infections were generally symptomatic, did the farmers select the fruit from the plants with the mildest symptoms to produce seeds? Or before the appearance of symptoms? Were certain CMV infections asymptomatic? Are there tolerance mechanisms specifically expressed in pepper fruits, whereby CMV infection leads to mild or no symptoms in fruits, despite severe symptoms in vegetative organs? These questions are still open.

## 5. Conclusions

The data presented here demonstrate that the widespread and recurrent CMV epidemics in the Espelette region are not explained by vertical (seed) transmission of the virus. In particular, year-to-year transmission of the virus does not rely on the vertical transmission of CMV from infected pepper plants to their progeny. Rather, in the presence of alternative hosts serving as viral reservoirs and inoculum sources in and around pepper fields, the horizontal transmission of the virus via aphid vectors is probably the main route for the initiation and spread of CMV epidemics in Espelette.

Interestingly, although our study produced no evidence of the seed transmission of CMV in Espelette pepper, this mode of virus transmission could play an important role in the initiation of epidemics and their recurrence from year to year in other plant hosts, particularly weeds [23].

Future investigations will aim to characterize the role of viral reservoirs in order to design appropriate control measures. In particular, the host range of CMV is, globally, the widest known for a plant virus; however, host range can vary between strains of the virus, and a single strain is likely to infect only a subset of all possible CMV host species. Through field sampling, we have already identified some weed species as hosts of the CMV strains in Espelette [22]. It will therefore be important to assess their role in the spread of CMV epidemics in pepper crops.

## Figures and Tables

**Figure 1 viruses-15-02159-f001:**
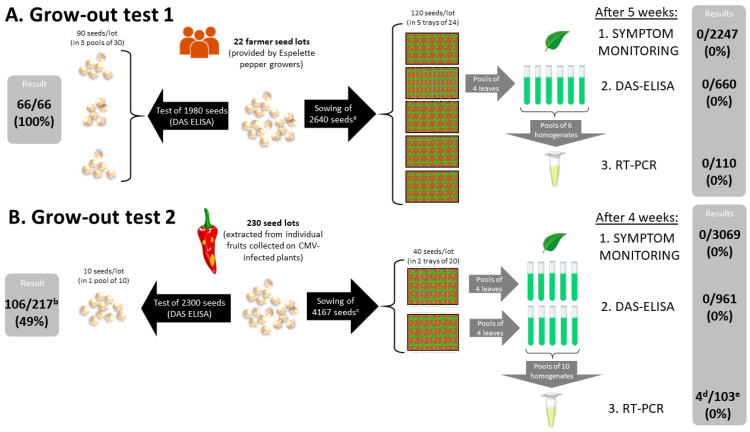
Overall scheme of the two grow-out tests. Pools of seeds, leaves, and homogenates may contain less than indicated, depending on the number of available seeds and seedlings. ^a^ In (**A**), the germination rate is 2247/2640 = 0.85; ^b^ only 217 fruits contained enough seeds to carry out the experiment. ^c^ in (**B**), only the 106 positive seed lots were sown, with a total of 4167 seeds (noting that not all lots contained 40 seeds). The germination rate is 3069/4167 = 0.74; ^d^ false positives due to cross contaminations (see Appendix A). The 113 seedlings corresponding to these 4 seed lots were tested again via DAS-ELISAs (113 individual tests) and RT-PCR (89 individual tests and two pools of 11 and 13 seedlings), and none of them were infected; ^e^ 3 seed lots did not yield any seedlings, resulting in only 103 seed lots being tested with RT-PCR.

**Figure 2 viruses-15-02159-f002:**
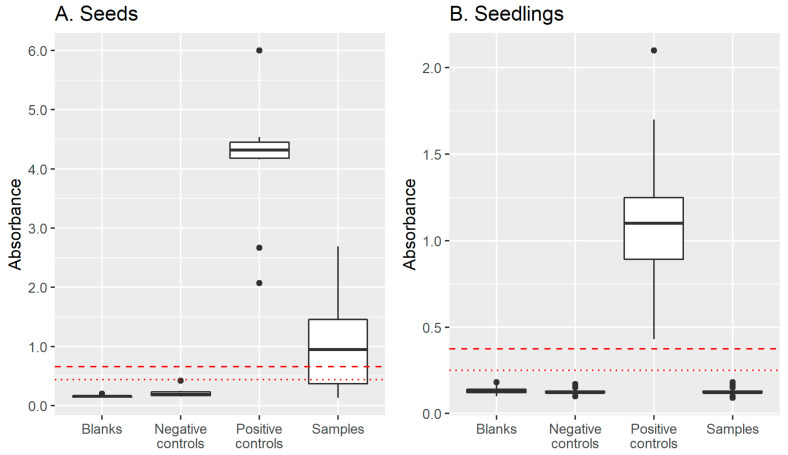
Results of DAS-ELISA tests on seeds and seedlings of grow-out tests 1 and 2. (**A**) For seeds, DAS-ELISAs were performed on 3 pools of 30 seeds per seed lot used in grow-out test one; a single pool of 10 seeds was tested for seed lots used in grow-out test two. Negative controls for seeds were non-infected Yolo Wonder (YW) pepper seeds. (**B**) For seedlings, assays were performed on pools of 1–4 leaves. Negative controls for leaf samples were non-infected YW leaves. For seed and leaf samples, positive controls were leaves of plants mechanically inoculated with a CMV isolate belonging to the main haplotype present on pepper crops in the Espelette area.

**Figure 3 viruses-15-02159-f003:**
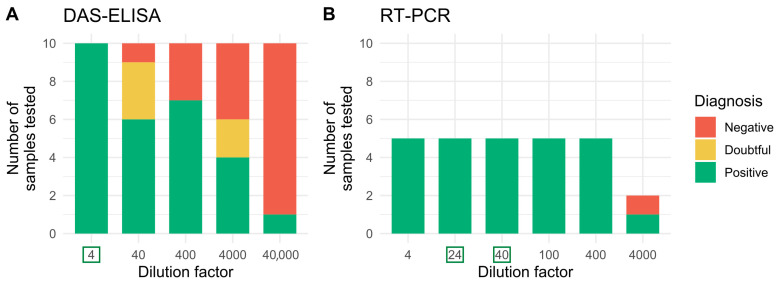
DAS-ELISA and RT-PCR sensitivity assays using pools of infected and healthy Gorria pepper leaves. The bar plots represent the number of samples that tested positive, doubtful, or negative for different dilution factors in the DAS-ELISA tests (**A**) or the RT-PCR tests (**B**). The dilution factors surrounded in a green box represent the size of the bulk samples used in the grow-out tests: 4 (i.e., 1 infected leaf mixed with 3 healthy leaves) for the DAS-ELISA tests, and 24 (1 infected/23 healthy) and 40 (1 infected/39 healthy) for RT-PCR in grow-out experiments one and two, respectively.

## Data Availability

The data presented in this study are available within the article or the Appendix A.

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
