# Peer review of "Absence of Seed-Mediated Transmission of Cucumber Mosaic Virus in Espelette Pepper Crops despite Widespread and Recurrent Epidemics"

_viruses, 2023, doi:10.3390/v15112159_

Round 1
Reviewer 1 Report
Comments and Suggestions for Authors In France, cucumber mosaic virus (CMV) have caused significant damage to Espelette pepper crops, and threatens the production of Espelette pepper. In present study, the authors determine whether CMV could be transmitted to seedlings from contaminated seeds, using visual symptom assessment, DAS-ELISA and RT-PCR. They found that the grow-out experiments on a total of over 5000 seedlings yielded no evidence of seed transmission of local CMV isolates in Gorria pepper. The results is very simple, clean and easy to understand. Following this assessment, I feel that the study can be accept as brief report or letter to editor after several revision in Viruses.1. How about other plant viruses occurred in Gorria pepper in France.
2. The authors speculated that ternative hosts present in and around pepper fields that can allow survival of CMV during the off season. I think the authors should give some evidence to support this speculation.
3. Several editorial errors were found in MS and references, the authors should modified the MS carefully again.
Comments on the Quality of English LanguageEnglish language of MS is considered available for "Viruses" Journal.
Reviewer 2 Report
Comments and Suggestions for Authors
The CMV reported in this paper is well documented in lots of previous studies. However, the capacity of seed transmission is till ambiguous. This paper clearly demonstrated that the CMV occurred in Espelette pepper did not transmitted by seed. It is a good case study for CMV seed transmission, so I recommand publishing this papper. And I have some suggestions to authors:
1. Do authors concern the sites of virus in the seed? The virus just only appeares on the outpeel of seed? Or probably in the embryo? I suggest the authors to dissect the seeds to detect the virus.
2. The neclutide sequece of CMV is the key evidence for stran identification, so I suggest putting the relevant evidences in the text.
Reviewer 3 Report
Comments and Suggestions for Authors
This is a valuable study that addresses important questions about the CMV epidemics in Espelette pepper crops. This study used multiple methods to confirm the presence or absence of seed transmission of CMV in pepper plants. Also, a higher number of replications further strengthens the confidence in the obtained results. Although the results obtained in this study are contrary to previous reports on seed/vertical transmission of CMV in pepper cultivars, the authors have done a good job of explaining the reasons for these discrepancies. I have minor comments below.
Line 44: Next-generation plants.
Line 44: undercover, greenhouse?
Line 91: remove indeed.
Line 120 infected-seeds
Line 120: grounded
Line 186: CMV is a non-enveloped virus. Why detergent was used for sterilization over bleach?
Line 213: This line is confusing, “after a 3-minute denaturation step of RNA and PCR reverse primer at 80°C, 213 followed by 5 minutes on ice” Please rephrase.
Line 25: collected from CMV
Line 219: remove, “which is 85% (farmers’ personal communication).”
Line 292: Please provide symptom observation last time point in the method section.
Line 347: remove, “in our experimental conditions”.
Line 355: 6 to 8 days after inoculation. This is very quick. Symptoms were observed on inoculated tissues or systemically infected leaves.
Line 367: Rephrase this line, it is kind of unattractive to the reader. “Indeed, in our laboratory experiments, the sensitivity assays of our ELISA and RT- 367 PCR showed that CMV from a single infected pepper leaf is still (and systematically) de- 368 tected even when mixed with 3 and 399 healthy leaves, respectively (Figure 3).”
Line 381: In the case of Tritimoviruses, Poaceviruses, and tobamoviruses, virus present in or on the seed coat or other tissues of maternal origin can be transmitted to the seedling through wounds that occur during germination or during transplanting or handling of seedlings (16,17, doi.org/10.3390/v15081774).
Comments on the Quality of English LanguageThe English language is fine. Minor editorial changes are suggested.
Round 2
Reviewer 1 Report
Comments and Suggestions for Authors
All my concens have been addressed. I recomend accept the MS.
Comments on the Quality of English LanguageThe quality of English in this paper is satisfied for the requirement of Journal.